# The Minus-End-Directed Kinesin OsDLK Shuttles to the Nucleus and Modulates the Expression of Cold-Box Factor 4

**DOI:** 10.3390/ijms23116291

**Published:** 2022-06-03

**Authors:** Xiaolu Xu, Sabine Hummel, Klaus Harter, Üner Kolukisaoglu, Michael Riemann, Peter Nick

**Affiliations:** 1Molecular Cell Biology, Botanical Institute, Karlsruhe Institute of Technology, Fritz-Haber-Weg 4, D-76131 Karlsruhe, Germany; michael.riemann@kit.edu; 2Center for Plant Molecular Biology (ZMBP), University of Tübingen, Auf der Morgenstelle 32, D-72076 Tübingen, Germany; sabine.hummel@zmbp.uni-tuebingen.de (S.H.); klaus.harter@zmbp.uni-tuebingen.de (K.H.); uener.kolukisaoglu@zmbp.uni-tuebingen.de (Ü.K.)

**Keywords:** Cold-Box Factor 4, cold stress, kinesin, microtubules, nuclear import, rice (*Oryza sativa* L.)

## Abstract

The transition to terrestrial plants was accompanied by a progressive loss of microtubule minus-end-directed dynein motors. Instead, the minus-end-directed class-XIV kinesins expanded considerably, likely related to novel functions. One of these motors, OsDLK (Dual Localisation Kinesin from rice), decorates cortical microtubules but moves into the nucleus in response to cold stress. This analysis of loss-of-function mutants in rice indicates that OsDLK participates in cell elongation during development. Since OsDLK harbours both a nuclear localisation signal and a putative leucin zipper, we asked whether the cold-induced import of OsDLK into the nucleus might correlate with specific DNA binding. Conducting a DPI-ELISA screen with recombinant OsDLKT (lacking the motor domain), we identified the Opaque2 motif as the most promising candidate. This motif is present in the promoter of *NtAvr9/Cf9*, the tobacco homologue of Cold-Box Factor 4, a transcription factor involved in cold adaptation. A comparative study revealed that the cold-induced accumulation of *NtAvr9/Cfp9* was specifically quelled in transgenic BY−2 cells overexpressing OsDLK-GFP. These findings are discussed as a working model, where, in response to cold stress, OsDLK partitions from cortical microtubules at the plasma membrane into the nucleus and specifically modulates the expression of genes involved in cold adaptation.

## 1. Introduction

The transition to a terrestrial lifestyle confronted plants with a fundamentally different situation where they had to develop self-supporting structures to compensate for the mechanic load, which was no longer carried by buoyancy. This transition left fundamental traces in the organisation and composition of the cytoskeleton (for a review, see [1]). One of the most striking changes was the progressive loss of dynein motors [2], while the specific class of the minus-end-directed class-XIV kinesins expanded [3], acquiring novel functions. For instance, the kinesins with a calponin-domain homologue link the mechanically rigid microtubules with the flexible actin filaments, which helps the pre-mitotic nucleus to locate the cell centre before initiating mitosis [4]. The kinesins KatA and ATK5 shorten the spindle during anaphase, thus adopting functions exerted by dyneins in animal cells [5,6].

In our previous work [7], we studied a rice homologue of KatA and ATK5, which turned out to occur at two sites in the cell and was, therefore, named the Dual Localisation Kinesin (DLK). While decorating cortical microtubules during the interphase and associating with different mitotic microtubule arrays during mitosis, it also moved into the nucleus in response to cold. Motility assays in vitro showed that this unconventional kinesin can drive the mutual sliding of microtubules and moves towards the minus-end of microtubules (i.e., in a direction typical for dynein motors) with a velocity comparable to other class-XIV kinesins. The accumulation of this kinesin in the nucleus could also be promoted in the absence of cold, when the cells were treated with Leptomycin B, a blocker of the nuclear export.

When a kinesin motor accumulates in the nucleus in response to a signal (cold) and is actively exported, there must be a functional reason, because the cytoskeleton is supposed to be tightly excluded from the karyoplasm during interphase. However, this textbook dogma has been progressively perforated during the last two decades. The seminal discovery of chromokinesin, a KIF4 kinesin constitutively localised in the mammalian nucleus [8], was later followed by findings where a KIF7 motor acts as a transcriptional regulator in the hedgehog signalling pathway [9]. Furthermore, in plants, a shuttling kinesin, OsBC12, was found to act as a transcription factor for the *ent*-kaurene oxidase OsKO2, such that the respective mutant is a gibberellin deficient dwarf [10].

This nuclear import of kinesins might be functionally linked to the nuclear import of tubulin that can be observed in plant cells in response to cold stress [11]. After the end of the cold period, tubulin rapidly leaves the nucleus again to build up a new cortical array of microtubules. Although tubulins lack canonical Nuclear Localisation Sequences (NLS), they are endowed with specific Nuclear Export Sequences (NES) that mediate the interaction with the Exportin receptor complex. These NES exist in plants as well and are functional as shown by constructs, where these sequences were placed in front of a GFP reporter [12]. While this export might be important to remove tubulin heterodimers that had been trapped during the formation of daughter nuclei in the telophase, the export inhibitor Leptomycin B can induce conspicuous accumulation of tubulin in the non-cycling stationary phase, which means that tubulin is also imported to a certain extent during the interphase, but is normally removed rapidly, such that the steady-state level of nuclear tubulin is low. Whether the intranuclear accumulation of tubulin in response to cold is caused by the higher sensitivity of this export as compared to the import or by the increased level of non-assembled tubulin from the eliminated microtubules remains to be elucidated.

When a kinesin motor, as well as its substrate, tubulin, partition to the nucleus in response to cold, there must be a functional relevance of this co-transport. The current work was motivated by the attempt to gain insight into this potential function. The analysis of the loss-of-function for DLK in rice and the patterns observed for the accumulation of the OsDLK transcript indicate that this kinesin, under normal conditions, acts in concert with cortical microtubule arrays required for cell elongation. The cold-induced nuclear import seems to function with a nuclear localisation signal, a putative leucin zipper, and the recognition of a specific DNA binding motif identified by a DPI-ELISA screen with N-terminal domain of recombinant OsDLK. The presence of this motif in the promoter of Cold-Box Factor 4, a transcription factor crucial for cold adaptation, along with the finding that the overexpression of OsDLK in tobacco specifically modulates the expression of this Cold Box Factor, point to a function of this unusual kinesin motor in the transduction of and adaptation to cold stress.

## 2. Results

### 2.1. OsDLK Is Essential for Early Development

To gain insight into the potential function of OsDLK in rice, two knock-out lines in which the gene was interrupted by either a T-DNA (FG_3A-07110.R) or by the rice retrotransposon Tos−17 (ND4501_0_508_1A) were selected from RiceGE and maintained over two additional generations of selfing as heterozygous populations. Then, seedlings were raised and genotyped, using two specific primer pairs. Among the 12 successfully genotyped individuals for the T-DNA insertion line, only 2 were assigned as WT, and only 1 as a homozygote. The vast majority (9 out of 12) were heterozygous (Appendix A). For the Tos−17 line, 6 out of 15 successfully genotyped lines were WT, while 9 were heterozygotes and none were homozygous (Appendix A). The very few homozygotes (HO) that could be recovered from genotyping never made it beyond the early seedling stage, indicating that homozygous *dlk* mutants are not viable. Thus, the lines had to be maintained by selfing the heterozygotes to screen the segregating population by genotyping.

To test for potential phenotypes in the heterozygotes, seedlings were raised in the dark for ten days to ensure that the coleoptile was fully elongated, even in the case of delays in germination. For the Tos−17 insertion lines (Figure 1A), the coleoptiles of the WT background (cultivar ‘Nipponbare’) reached 33 mm on average, while in the heterozygotes, they were slightly (by 3 mm), but significantly, shorter. Among the germinated seeds, not a single homozygote was found. For the T-DNA insertion (Figure 1B), the coleoptiles of the WT background (cultivar ‘Dongjin’) reached approximately 22 mm on average, while coleoptiles of the heterozygotes reached only 15 mm. Here, a few homozygotes germinated, but coleoptile elongation arrested at approximately 3 mm. Thus, there was a mild, but significant, reduction of coleoptile elongation in the heterozygotes, irrespective of the type of insertion and WT background.

The phenotype of the heterozygotes indicated a role of OsDLK in coleoptile elongation. We therefore measured steady-state transcript levels in ‘Nipponbare’ (the WT background for the Tos−17 insertion line) from the first day after sowing until day 6, when coleoptiles stopped growing and opened (Figure 1C). Transcripts were found to start at a high level until day 4, when they declined rapidly and, from one day later, remained at approximately half of the initial level. This decline correlated with the pattern of coleoptile growth (which is typical not only for ‘Nipponbare’, but for *japonica* varieties in general): after a lag phase of around 2 days, the emerging coleoptiles grew rapidly over the subsequent two days, and then slowed down after day four. Following day 5, hardly any length increment was observable, and from day 6, coleoptile opened along their preformed ventral seam. Thus, the steady-state levels of OsDLK transcripts were high prior to and during rapid elongation, and their decrease preceded a decrease in elongation. This expression pattern is consistent with the phenotype of the insertion mutants, where homozygotes are severely impaired in germination, and even heterozygotes show moderate but significant reductions of coleoptile elongation, indicating that the expression of OsDLK becomes limiting during that stage. The decrease in the transcript prior to the boost of elongation rate is consistent with the scenario where these transcripts are consumed for elongation.

### 2.2. OsDLK Is Upregulated in Elongating Tissues

To gain insight into the role of OsDLK during early development, seedlings were grown under light for ten days, and then the steady-state transcript levels of OsDLK in different organs were measured by real-time qPCR (Figure 2A). The highest transcript levels were found in the sheaths of the second leaf that was rapidly elongating during that stage, and the third leaf that was in the process of unfolding. In contrast, expression in the first leaf, which, at that stage, had already stopped elongating, was low, as was that in the blade of the second leaf, which was already fully expanded. The expression in the seminal root and crown roots was also relatively low. Thus, the expression of OsDLK reflects the pattern of organ expansion. We also phenotyped the (heterozygous as no homozygotes survived the seedling stage for the T-DNA nor the Tos−17 rice lines) development of the second leaf sheath in light-grown plantlets at day 10 after germination (Appendix A). We found that the second leaf sheath of the heterozygotes was shorter than in the wildtype plants. This difference did not reach significance in the case of the Tos−17 mutant, but was significant at *p* < 0.01 in the case of the T-DNA mutant.

Since organ expansion is controlled by auxin, especially in coleoptiles, we probed for the potential induction of OsDLK by auxin using the classical coleoptile segment assay. Coleoptile segments of fixed lengths were excised, the endogenous auxins were washed out, and the segments were then incubated in different concentrations of exogenous indole acetic acid (IAA). Expression was scored 1 h later to determine the immediate response, and 6 h later to assess whether this response was transient (Figure 2B). To monitor the activity of auxin signalling, we measured steady-state transcript levels of the auxin-response factor OsIAA9-2. The transcripts for this marker at 1 h increased steadily from 0.1 µM IAA, reaching more than a 20-fold induction for the highest tested concentration of 100 µM IAA. At 6 h, the dose–response curve shifted to the right by more than one order of magnitude, the threshold was higher than 1 µM IAA, while the slope of the curve was comparable to that seen at 1 h after the addition of IAA. This indicates a decrease in IAA sensitivity (Figure 2B), because, to reach a certain response, more auxin is needed if scored at 6 h as compared to 1 h. Whether this drop in sensitivity is caused by a reduced abundance of auxin receptors or reduced activity of receptors remains to be elucidated, but the down-stream signalling seems unaltered, since the slope of the two curves is the same. The upregulation of OsDLK transcripts became detectable from 1 µM of IAA and reached a plateau from 10 µM of IAA, albeit the increase in steady-state levels was very moderate (by around 25%) and significant only for the highest tested concentration of IAA (100 µM). For 6 h of incubation, there was a clear and significant down-regulation of approximately 50% saturated from 10 µM of IAA. Thus, for 1 µM IAA, the OsDLK transcript is rapidly but transiently up-regulated and then returns to the resting level, while for higher concentrations, up-regulation is followed by significant down-regulation. This dose-dependency mirrors the dose-dependency seen in rice coleoptile segments for the auxin-inducible auxin response factor OsARF1 [13], but also that of elongation growth [14].

### 2.3. In Response to Cold, OsDLK-GFP and Microtubules Dissociate

The dual localisation of an OsDLK-GFP fusion had been shown for interphase cells in *Arabidopsis thaliana* protoplasts and BY−2 cells, as well as for epidermal cells of the rice leaf sheath [7]. For tobacco cells, the population of OsDLK-GFP that was bound to cortical microtubules vanished, while the signal in the nucleus increased concomitantly. To test whether this release of tagged OsDLK from cortical microtubules was also preserved in the homologous model system, rice, etiolated coleoptiles were transiently transformed by biolistics and then, upon expression of the OsDLK-fusion in the hit epidermal cells, kept on ice. Prior to the cold stress, the nucleus was labelled by GFP along with punctate structures along the cell periphery that were aligned similar to beads on a string (Figure 3A,B). Magnification (Figure 3D) showed that these punctate structures were connected by microtubule-like structures that were aligned in a steeply oblique orientation typical for epidermal cells grown under a light. Two hours after the onset of cold stress, the punctate structures persisted, but their microtubular connections had completely vanished (Figure 3C,E). Interestingly, the position of the nucleus had shifted significantly in direction to the pole (compare Figure 3B,C), concomitantly with the elimination of cortical microtubules. Thus, in the homologous system of rice, OsDLK-GFP is rapidly released from microtubules in the response to cold.

### 2.4. The Cold Response of the Tobacco CBF4 Homologue Is Quelled by OsDLK Overexpression

To gain insight into the potential functions of OsDLK, we assessed the expression of a panel of cold-stress-related genes in non-transformed tobacco cells and in cells overexpressing OsDLK as a fusion with GFP [7]. We tested NtCf9, the tobacco homologues of Cold-Box Factor 4 (Appendix A), a key regulator of cold acclimation [15]; tobacco Cold Box Factor 2 (NtCBF2), a transcriptional activator of cold-responsive genes [16]; tobacco Inducer of CBF expression (NtICE2), the master switch for CBFs [16]; tobacco Late Elongated Hypocotyl (NtLHY), a regulator of CBFs acting in parallel to ICE [16]; tobacco Gigantea (NtGIGANTEA), a positive regulator of freezing tolerance acting independently of CBFs [17]; tobacco Timing of Cab Expression 1 (NtTOC1), a phytochrome-dependent repressor of CBF expression [18]; tobacco Hypocotyl 5 (NtHY5), a light-dependent regulator of cold acclimation acting independently of CBFs [16]; tobacco Early Flowering 3 (NtELF3), a phytochrome-dependent regulator of CBFs [18]; and tobacco High Expression of Osmotically Responsive Genes (NtHOS1), a negative regulator of CBFs [19]. With the exception of NtCf9 and NtCBF2, there was no significant induction of any of the tested transcripts (Appendix A and Figure 4A). NtCBF2 expression increased extraordinarily by approximately four orders of magnitude (Appendix A), equally in non-transformed cells and cells overexpressing the OsDLK-GFP fusion. However, the expression of NtCf9, which was strongly induced in both cell lines, nevertheless showed a clear difference. Compared to the non-transformed cells, it accumulated much slower and only at 30% amplitude in the cells overexpressing OsDLK-GFP (Figure 4B). This difference is specific to the NtCf9 transcript and indicates negative regulation by the introduced transgene.

### 2.5. OsDLK Shows Affinity to Different DNA Motifs

Detailed sequence analysis of OsDLK shows, in addition to the features characteristic for minus-end-directed kinesin motors (such as a C-terminal head domain harbouring ATP- and microtubule-binding sites and a neck region with a characteristic signature for inverse directionality), motifs indicating a nuclear function. These are localised in the non-motor N-terminal domain of the protein (Figure 5A) and include a predicted Nuclear Localisation Signature and a Leucine Zipper motif predicted to mediate nucleotide binding. In addition, there are six coiled-coil domains indicating potential interaction with other partners.

To test whether this N-terminal domain can bind DNA specifically, we expressed the N-terminal part containing the tail domain (1−1209 bp, DLK−Δ_1−403_) of OsDLK as His-fusions to explore the DNA–protein interaction (DPI) with an anti-penta His antibody as the detecting agent. We were able to express the construct in a soluble manner, and the anti-penta His antibody detected specific bands. For the truncated DLK−Δ_1−403_, a dominant band at approximately 50 kDa could be detected, consistent with the expected size for a penta-His fusion in the predicted 47.3 kDa N-terminal domain (Figure 5B), along with a smaller fragment of around 40 kDa, which might represent a proteolytic cleavage product.

In a DPI-ELISA screen (Figure 5C), which is based on a library of double-stranded DNA probes that are coupled to the surface of microtiter plates for high-throughput screening of hexanucleotide motifs [20], the DNA binding properties of the recombinant proteins were tested, and were able to find six motifs that were bound by the truncated, N-terminal half of the protein DLK−Δ_1–403_ (Table 1). Five of these motifs corresponded to cis elements known from the literature and related to predicting functions such as salicylic acid responsiveness, anaerobic induction, or endosperm expression. Within this set of candidates, motif Nr. 294 was the most promising. This motif aptly matched, over a stretch of 9 bp, the O2 motif shown to be involved in the metabolism regulation of the maize seed protein zein. Moreover, this oligo motif Nr. 294 was present in the promoter of NtCf9 (779−788 bp before the start codon, Figure 5D), i.e., in the gene, whose induction with cold was specifically quelled upon the overexpression of OsDLK in tobacco (Figure 4B).

## 3. Discussion

The current study tried to gain insight into the biological function of the unusual kinesin motor Dual Localisation Kinesin (DLK), by analysing the phenotype for rice loss-of-function mutants, the regulatory pattern in rice, and the effect of overexpression in tobacco BY−2 cells. Since the localisation of this protein depends on temperature, one needs to consider its role under normal temperature separately from its function under cold stress. In fact, our data are compatible with a model where DLK under normal conditions sustains cell elongation by fulfilling a function associated with the cortical microtubule. However, in response to cold stress, this kinesin enters the nucleus, and binds specifically to target motifs in the DNA. As a result, the induction of transcripts for Cold-Box Factor 4 in response to cold is modulated, indicating a function of this unusual kinesin motor in the transduction of and adaptation to cold stress. These findings stimulate the following questions to be discussed: 1. What is the potential link between DLK and cell elongation? 2. What is the molecular base for the transcriptional activation? 3. How is the nuclear import of DLK connected to cold signalling and cold acclimation?

OsDLK and elongation. Under normal temperatures, OsDLK prevails at the cortical microtubules, which holds true, both for heterologous ([7] tobacco BY−2, cotyledons of *Arabidopsis thaliana*), as well as homologous, systems (Figure 3: Rice coleoptile epidermis). This localisation indicates a function for cell elongation. This is supported by the finding that loss-of-function mutants (T-DNA, Tos−17 retrotransposon) show impaired elongation in the heterozygous state (Figure 1A,B). In the homozygous state, they are very short (T-DNA, Figure 1B) or even non-viable (Tos−17 retrotransposon, Figure 1A). The fact that OsDLK is necessary for coleoptile elongation is further corroborated by the temporal pattern of OsDLK transcripts in etiolated coleoptiles (that grow exclusively via cell elongation). Here, the steady-state transcript levels are high up to day 4 after sowing and then decrease substantially during the phase of logarithmic expansion (Figure 1C). As to be expected, the expression of OsDLK is strong in rapidly expanding parts of the leaves (second sheath, third leaf), while it is low in the roots, where growth proceeds concomitantly with mitosis (Figure 2A). Conversely, the transient induction of OsDLK transcripts by the natural auxin indole-acetic acid is consistent with a role for OsDLK in elongation (Figure 2B).

These considerations lead to the question by which mechanism can OsDLK sustain cell elongation. Coleoptiles grow exclusively by cell elongation without proliferation, and this expansion is constrained and directed by the mechanic properties of the outer epidermal cell wall [21]. Elongation is sustained by a transverse orientation of newly deposited cellulose microfibrils that reinforce the otherwise preferentially transverse stress from the expanding protoplast [22]. When microtubules re-orient into more longitudinal arrays in response to phototropic or gravitropic, this is followed by a decrease in the elongation rate and in one flank of the coleoptile, culminating in tropistic bending [23]. In *Arabidopsis thaliana*, the class-XIV kinesin ATK5 (the closest homologue of rice DLK) is not only found in spindles and phragmoplasts, but also decorates cortical microtubules [6]. This protein has been shown to exert a bundling activity in the spindle. So, it is straightforward to assume that ATK5 (as well as its rice homologue DLK) will do so in the cortical array, thus stabilising a transverse orientation of microtubules and, in this way, sustaining coleoptile elongation.

OsDLK and transcriptional regulation. We find that the overexpression of OsDLK specifically modulates the expression of *NtCf9*, the tobacco homologue of Cold-Box Factor 4 (Figure 4B), while the responses of other cold-related transcripts did not show a significant difference compared to non-transformed tobacco cells (Figure 4A). Recombinantly expressed OsDLK bound to a specific DNA motif, 11 bp in length (Figure 5B–D), which has been known to be a target for transcription factor OPAQUE 2, a transcriptional key regulator that activates the accumulation of seed storage proteins by binding to this motif by virtue of its leucin zipper [24,25]. Interestingly, there exist several functional analogies between OPAQUE 2 and OsDLK.

OPAQUE2 regulates zein genes and has been shown to move into the nucleus after ubiquitination through the E3 ubiquitin ligase RFWD2 [26]. The sucrose-responsive kinase SnRK1 phosphorylates RFWD2 and initiates its degradation, which is a mechanism to regulate the accumulation of zein depending on sucrose. SnRK1, in turn, is negatively regulated by the phosphatases ABI1 and PP2CA [27]. At this point, there is an interesting bifurcation: When these phosphatases are activated, this is also a release of the kinase Open Stomata 1 from the membrane, such that it can activate the master switch ICE1, governing the cold-induced transcriptional cascade in the nucleus (for a review, see [28]). The activation of ABI1 and PPC2A by cold stress would, thus, activate SnRK1, such that RFWD2 is phosphorylated and removed and OPAQUE2 would not move into the nucleus. Under the same conditions, OsDLK would move into the nucleus. Thus, both proteins not only share motifs in the promoters of their targets, but also move into the nucleus depending on signals, and they share upstream elements of signalling. However, their behaviour in response to cold stress is expected to be a mirror image.

That a kinesin function is physically linked to a function in transcriptional regulation appears exotic at first, but OsDLK is not the only case for such a link: For instance, the KIF7 motor regulates hedgehog signalling in mammalian cells [9], and the human class-XIV motor KIFC1 was shown recently to be required for DNA replication during the S-phase [29]. Likewise, the human motor KIF4 is localised in the nucleus throughout the interphase and participates in chromatin remodelling [30]. In plants, so far, there is only one further case known where a kinesin exerts a function in DNA regulation, namely the rice kinesin OsBC12 activating expression of the gibberellin synthesis gene *ent*-kaurene oxidase [10].

Are nuclear kinesins rudiments of evolution? Microtubules and DNA are strictly sequestered during the interphase, separated by the nuclear envelope that tightly controls which molecules are allowed to pass through nuclear pores (for a classical review, see [31]). The directionality is maintained by a gradient of the small GTPase Ran, which is found as Ran-GTP in the karyoplasm but as Ran-GDP in the cytoplasm. These forms of Ran are maintained by accessory proteins, such as RanGAP, which, in plants, is located at the outer nuclear envelope [32] Ran-GTP can induce microtubule nucleation and initiate spindle formation. However, this is prevented during the interphase because tubulin is strictly separated from Ran-GTP. When the nuclear envelope breaks down at the onset of the metaphase, Ran-GTP can initiate microtubules in the former karyoplasm. In fact, plant RanGAP associates with tubulin in mitotic cells, but not in cells that are not cycling [32]. Tubulin is void of a canonical NLS, but it does harbour functional NES motifs [12] indicating that the exclusion of tubulin from the interphasic karyoplasm is essential. The strict sequestering of chromatin from tubulin is not a primordial phenomenon, but evolved from an ancestral situation, where tubulin was in the nucleus and only later “explored” the surrounding cytoplasm (for a review, see [33]). Numerous transitional stages, where the spindle can develop without the breakdown of the nuclear envelope, can be considered evolutionary witnesses of this shift out of the nucleus. In response to cold stress, tubulin can return swiftly to this evolutionarily ancient state and accumulate in the nucleus, from where, upon rewarming, it rapidly returns to the cytoplasm, organising new microtubular structures [11]. Since tubulin dimers are approximately twice as large as the size exclusion limit of nuclear pores, they would be expected to require an NLS in doing so. Since, so far, no NLS could be identified in any tubulin, it is more likely that tubulin enters the nucleus as part of a complex. Whether DLK is a component of this complex is not known but represents an attractive hypothesis that should be tested in future experiments.

Is DLK a negative regulator of cellular cold acclimation? Low temperature is among the major environmental factors shaping the geographical distribution of plants and can deploy a plethora of damage-related and adaptive responses, depending on the species, the harshness of the condition, and the time course of the stress (classical and comprehensive reviews that are still worth reading are those by Lyons and Burke [34,35]). Not surprisingly, the plant responses to this environmental complexity are complex and specific as well. Many plants can deploy efficient cellular adaptation to even harsh cold stress if this stress is preceded by prolonged but mild chilling, a phenomenon known as cold acclimation or cold hardening. The inducing effect of this chilling pre-treatment can be phenocopied by a transient elimination of microtubules (winter wheat [36] and grapevine cells [37]). Cold hardening is accompanied by the high expression of Cold-Box Factor 4 (grapevine plants [38] and grapevine cells [37]), and the overexpression of Cold-Box Factor 4 can induce freezing tolerance in grapevine without the need for pre-chilling [39]. Since the perception of cold occurs at the plasma membrane, while the activation of transcriptional regulators proceeds in the nucleus, signals must be conveyed from the membrane to the nucleus. In fact, MAPK cascades are deployed in response to cold stress and these cascades modulate the stability of transcriptional regulators in a complex, partially antagonistic manner. In parallel, other signals, such as 14−3−3 proteins [40] or the kinase Open Stomata 1 [41] can, in response to cold stress, travel to the nucleus and modulate the activity of these transcriptional regulators. While this signalling appears redundant at first, it can be functionally dissected depending on time—some of these signals (such as Open Stomata 1) are rapid and might be elements of signal transduction, while others (such as inhibitory activities of MAP kinases) occur later and might be elements of signal habituation (for a review, see [28]). The multitude of events linked to membrane–nucleus crosstalk leads to the question of how the nuclear import of DLK integrates with this complex network and what the functional relevance of this phenomenon might be.

A slow and steady drop in temperature, as typically occurs in autumn, allows for cellular adaptation, for instance, by producing proteins that protect membranes and organelles from ice damage as would otherwise occur later in the winter. The accumulation of these cold-responsive (COR) proteins is under the control of the CBF-transcriptional cascade (for a classical review, see [42]), and CBF4 as a slower, but a sustained member of this cascade might be the pivotal driver of long-term acclimation. However, the response to a rapid cold snap must be different because there is no time to bring cold acclimation to a successful end. In contrast, the plant will only have a chance to survive if it succeeds in mobilising resources from older and more dispensable organs towards meristematic tissues, from where new organs can be re-generated once the cold-stress episode has faded [43]. This scenario would require cold-induced programmed cell death (PCD), a process that has, indeed, been demonstrated for tobacco BY−2 cells [44]. To initiate CBF4-dependent cold acclimation under these conditions would be counterproductive. Thus, it is expected that, under conditions of harsh and acute cold stress, the induction of CBF4 that would otherwise ensue from the response to the accumulation of ICE1, has to be suppressed. DLK, as a negative regulator that, in response to such harsh cold stress, is released from the disassembled microtubules (and thus, also will expose its DNA binding leucin zipper in the N-terminal domain), might act as a switch that quells cold acclimation under conditions where cells undergo cold-induced PCD. An implication of this working hypothesis is that the overexpression of CBF4 should mitigate cold-induced mortality. In fact, this is what we see in ongoing experiments.

## 4. Materials and Methods

Database search for cDNA clones and knock-out mutants. The genomic sequence of OsDLK (accession number Os07g01490) was screened for the availability of knock-out mutants in the RiceGE database (http://signal.salk.edu/cgi-bin/RiceGE, accessed on 18 October 2013) leading to the identification of the T-DNA insertion line PFG_3A-07110.R (in the background of the *japonica* cultivar ‘Dongjin’), and the Tos-17-insertion line ND4501_0_508_1A (in the background of the *japonica* cultivar ‘Nipponbare’, NIAS, Tsukuba, JapanDongjinPFG_3A-07110.ROryza sativa L. japonica cv. Dongjin, T-DNA insertion line; Postech, South Korea). Seed material for the T-DNA insertion lines was kindly provided by the National Research Laboratory of Plant Functional Genomics, Division of Molecular and Life Sciences, Pohang University of Science and Technology (POSTECH), Pohang 790−784, Korea, while material for the Tos−17 insertion line was from the National Institute of Agricultural Science (NIAS), Tsukuba, Japan.

Cultivation of rice. For genotyping, the sterilised seeds were grown on floating meshes in photo-biological darkness (using boxes wrapped in black cloth) at 25 °C for 4 days, as described by Nick et al. [45]. The etiolated coleoptiles were excised and immediately frozen in liquid nitrogen and stored at −80 °C until isolation. For the segregation analysis of coleoptile elongation, plants were raised in the dark, to compare the genotyped homozygotes or heterozygotes from the T-DNA or the Tos−17 insertion lines with the respective wildtype background, as well as with the segregating wildtype seedlings. Seedlings were digitalised on a scanner, and then the coleoptile length from the node to the tip was measured using the periphery tool of ImageJ (https://imagej.nih.gov/ij/, accessed on 13 April 2015). For the analysis of tissue regulation, seedlings were grown in Magenta boxes (Sigma-Aldrich, St. Louis, MO, USA) on 0.4% [*w*/*v*] phytoagar (0.6% *w*/*v*, Duchefa, Haarlem, The Netherlands) under sterile conditions as described in Tang et al. [46], either at 25 °C in darkness or under continuous white light.

Genotyping of rice mutants. Etiolated coleoptiles were shock-frozen in liquid nitrogen, and ground to a powder in a TissueLyser (Qiagen, Hilden, Germany). Genomic DNA was isolated from 100−200 mg of powdered material based on the standard cetyl trimethyl ammonium bromide (CTAB) protocol using chloroform-isopropanol for partitioning [47]. Diagnostic sequences for the genotype of the Tos−17 or T-DNA insertion mutants were amplified by two complementary polymerase chain reactions [48]. The first pair probed the flanking sequences of the putative insertion site, i.e., motifs that would be found in both wildtype and transformant (however, in the transformant, the amplicon would be larger), while the second pair probed a target site located in the respective insert (T-DNA or Tos−17, respectively), while the second site targeted one flanking region of the putative insertion. This reaction would only be amplified in the case of a transformation event. In the segregated wildtype (SeWT) of the T-DNA insertion line, the primer pair KinF1/R1 produced a fragment of 330 bp, while the transgenic allele did not lead to an amplicon. In the case of amplification with the same upstream primer (KinF1) and the T-DNA-specific downstream primer TR, a fragment of 592 bp was amplified for the transgenic allele, while the wildtype allele did not produce a product. Those plants that showed bands with both primers KinF1/R1 and KinF1/TR (flanking T-DNA insertion site) could, thus, be defined as heterozygotes (HZ), while in homozygotes (HO), only the band at 592 bp should be detected. The Tos−17 line was genotyped in a similar manner. In heterozygotes, both primers KinF2/R2 and KinF2/Tos17R would produce bands at 350 bp and 546 bp, respectively, while KinF2/R2 should work for SeWT plants only and KinF2/Tos17R for HO plants only. The sequences of primers for different mutants are listed in Appendix A, independently. All PCR reactions used Taq DNA polymerase (NEB) and the following conditions: Initial denaturation for 5 min at 95 °C, followed by 42 cycles of denaturation for 30 s at 95 °C; annealing for 30 s at 60 °C; synthesis for 30 s at 72 °C. The PCR products were separated on 1% [*w*/*v*] agarose gels.

Transient transformation of rice seedlings by biolistics. Seedlings of rice (*Oryza sativa* L. *japonica* cultivar *Nihonmasari*) were raised in darkness at 25 °C for 4 days. The rice coleoptiles were transiently transformed with the recombinant plasmid pK7FWG2-OsDLK in which OsDLK was C-terminally fused with GFP, via gold particle bombardment as described by Holweg et al. [49]. During biolistic transformation, the coleoptiles were arranged in the middle of PetriSlides (Millipore, Schwalbach, Germany) with 0.4% phytoagar and fixed with a wire grid. For each plasmid solution, three Petri dishes were prepared. The Petri dishes were then placed in the particle gun and were bombarded three times at a pressure of 2.5 bar in the vacuum chamber at −0.8 bar. Following bombardment, the transformed rice coleoptiles were returned to the dark at 25 °C for 24 h and transferred into an ice bath for the chilling assay.

Microscopy and image analysis. Following biolistic transformation and incubation for 24 h in the dark for the expression of the transgene, individual cells of the coleoptiles were followed over time during cold stress using an AxioImager Z.1 microscope (Zeiss, Jena, Germany) equipped with an ApoTome microscope slider for optical sectioning and a cooled digital CCD camera (AxioCam MRm). GFP fluorescence was recorded through the filter set at 38 HE (Zeiss; excitation at 470 nm, beamsplitter at 495 nm and emission at 525 nm). Acquired images were operated via the AxioVision (Rel. 4.8.2) software. The coleoptiles were excised from the seeds, split longitudinally, and then placed with their epidermal side positioned towards the objective and covered with coverslips. After collecting the z-stack, the slide with a coleoptile slice was returned to the ice bath for further incubation. The whole process was conducted in a dark room.

RNA isolation. The plant material was harvested into liquid nitrogen and kept at −80 °C until RNA extraction. The samples were ground to a powder using the TissueLyser (Qiagen, Hilden, Germany), and RNA was extracted from aliquots of 100 mg of these powdered samples using the innuPREP plant RNA kit (Analytik Jena, Germany), combined with RNase-free DNAse I (Qiagen) to avoid contamination with genomic DNA, according to the manufacturer’s instructions. The purity and integrity of the extracted RNA were determined both spectrophotometrically (Nano-Drop 2000) and by gel electrophoresis (on 1% [*w*/*v*] agarose gels).

Measuring steady-state levels of OsDLK transcripts in rice seedlings. The expression patterns of OsDLK were assessed in the O. sativa ssp. japonica cv. Nipponbare wildtype. To address the developmental time course of steady-state levels for the OsDLK transcript in rice, RNA was extracted at daily intervals from seedlings raised for up to 6 days at 25 °C under continuous white light (120 µmol·m^−2^·s^−1^ of photosynthetically available radiation; Neo tube TLD 36 W/25, Philips, Hamburg). For specimens that still had not initiated germination, the mature embryo was dissected, and for specimens that had germinated, the caryopsis was separated from the seedling. At least three entire individuals were pooled to extract RNA for each time point and experimental replication. To investigate the tissue specificity of OsDLK expression, the rice seedlings were grown under the same conditions for 10 days. Then, different organs, including the entire first leaf (having terminated growth at this stage), the blade and sheath of the second leaf (still elongating), the entire third leaf (in the early phase of elongation), the seminal root (having terminated growth at this stage), and the crown root (beginning to elongate), were separated carefully. Tissues for at least 12 individual seedlings were pooled for each data point and replication. For the auxin dose–response curve of DLK expression, segments of 10 mm length were excised under green safelight from etiolated coleoptiles grown for 4 days at 25 °C as described by [50]. The young leaf inside each coleoptile was removed with tweezers. The segments were first incubated in water for 1 h to deplete them from endogenous auxin. Subsequently, they were transferred to different concentrations of indole-3-acetic acid (IAA) and incubated in a shaker for either 1 h or 6 h. At least ten segments per data point and replication were pooled to extract RNA extraction and quantitative analysis. All data on transcript levels represent mean values and standard errors from at least three independent experimental series.

Steady-state transcript levels were measured by a quantitative Real-time polymerase chain reaction (RT-PCR) using GoTaq Polymerase (NEB) with an initial denaturation for 3 min at 95 °C, followed by 39 cycles of denaturation for 15 s at 95 °C, annealing for 40 s at 60 °C, and synthesis for 30 s at 72 °C, using a Bio-Rad CFX detection System (Bio-Rad, München, Germany) according to the manufacturer instructions. Signals were visualised using the iQ SYBR Green Supermix (Bio-Rad). The primer pair qDLK fw/re was used to detect the OsDLK transcripts level, and the primer pairs Ubiquitin-10 fw/re and GAPDH fw/re were used for the endogenous control gene (Appendix A).

Cold response gene expression in transgenic BY−2 cells. The cold response of gene expression was mapped in tobacco BY−2 (*Nicotiana tabacum* L. cv. *Bright Yellow 2*) cells that were either non-transformed (wildtype) or that overexpressed a GFP fusion of OsDLK [7]. Both cell lines were cultivated at 25 °C until the end of their cycling phase (day 3 after subcultivation). To administer cold stress, the flasks with cells were transferred into an ice bath on an orbital shaker at 100 rpm for up to additional four days. In addition to the foreign OsDLK transcripts, several cold-responsive genes of tobacco itself were followed over time. The details for the respective oligonucleotide primers are given in Appendix A. The ribosomal gene L25 and elongation factor gene EF-1α were used as housekeeping genes for normalisation. Quantitative RT-PCR was conducted using GoTaq Polymerase (NEB) with an initial denaturation for 3 min at 95 °C, followed by 39 cycles of denaturation for 15 s at 95 °C, annealing for 40 s at 58 °C, and synthesis for 30 s at 72 °C.

Recombinant protein expression and Western Blot. To screen protein–DNA interactions, OsDLKT, a partial fragment containing the tail part harbouring the motor domain of OsDLK, was cloned into the binary plasmid pET-DEST42 via GATEWAY^®^ cloning as described by Klotz and Nick [51]. The partial construct OsDLKT was amplified using the forward primer:

5′-GGGGACAAGTTTGTACAAAAAAGCAGGCTTCATGTCCACGCGCGCCACTCGCC-3′

and the reverse primer:

5′-GGGGACCACTTTGTACAAGAAAGCTGGGTCATTCTCTCCGTCCAAAATTTGTT-3′.

The recombinant plasmid pET-DEST42-OsDLKT was transferred into the *E. coli* strain BL21-Codon Plus (DE3)-RIL. Protein expression was induced by 200 µM of isopropyl-β-D-thiogalactopyranoside (IPTG) in culture flasks containing LB medium supplemented with ampicillin, at 37 °C for 4 h. Cells were collected by centrifugation at 4500× *g* at 4 °C for 20 min (Hermle Universal centrifuge, Wehingen, Germany), and the sediments were washed with DPI-ELISA buffer (4 mM HEPES pH 7.5, 100 mM KCl, 8% glycerol), supplemented with 1 mM phenyl-methyl-sulphonyl-fluoride (PMSF) and a proteinase inhibitor cocktail (Roche, Germany). The cells were lysed by sonication (UP100H, Hielscher, Germany) with a frequency of 6 cycles of 15 s with 15 s interruption, at 80% power. The crude extracts for proteins of interest were collected through centrifugation at 4500× *g* at 4 °C for 20 min. As a negative control, cells transformed by the empty vector pET-DEST42 were included as well. The crude extract samples were mixed with the sample buffer. After heating at 95 °C for 5 min, the mixture was split equally into two SDS-polyacrylamide gels and run at 25 mA for 90 min. One of the gels was stained for recording the loading, while the lanes of the other gel were electroblotted onto a polyvinylidene fluoride (PVDF) membrane (Pall Gelman Laboratory, Dreieich, Germany). The expression of the protein was detected with the mouse monoclonal antibody anti-penta His (Qiagen, Hilden; Germany) in a 2000-fold dilution of Tris-Buffered Saline with 0.1% Tween^®^-20 (TBST). The primary antibody was visualised using anti-mouse IgG conjugated with alkaline phosphatase in a dilution of 1:50,000 of TBST. The signal was developed using the Alkaline Phosphatase Color Development Kit (Thermo Scientific, Langenselbold, Germany).

Identification of DNA binding motifs. To identify candidates for DNA-binding targets of OsDLKT, a DNA–protein-interaction (DPI)-ELISA strategy was used [20] based on an optimised double-stranded DNA (dsDNA) probe library that allows the high-throughput identification of hexa-nucleotide DNA-binding motifs. The presence of putative *cis* elements in potential target promoters was predicted via Plant Care (http://bioinformatics.psb.ugent.be/webtools/plantcare/html/, accessed on 6 April 2017).

## 5. Conclusions

Our working model proposes that the dual localisation of DLK is a reflection of a dual function, linking cold perception at the membrane with adaptive (CBF4) or self-destructive (suppression of CBF4) responses. This assigns DLK the role of a switch that will either secure the survival of the cell (in the case of cold acclimation) or its controlled decay, such that energy resources can be mobilised for the sake of other cells (e.g., in the meristem) that will survive and restore the integrity of the organism. If this working model held true, one would predict that the expression of cold-responsive genes should depend on the type of cold stress (harsh versus milder chilling), because these affect the cortical microtubules to a different degree. Moreover, elements of signal transduction that are independent of microtubules and DLK, for instance, 14−3−3 proteins [40], would contribute differently to the expression of such cold-responsive genes depending on the stringency of the cold stress. Whether DLK detaches from the disassembling microtubules, or whether it is released because microtubules just disappear, is not clear at the moment. Hence, it would be interesting to suppress microtubule disassembly in the cold by pre-treatment with taxol and determine whether DLK will move into the nucleus anyway. Last, but not least, it would be worthwhile to engineer truncated versions of DLK, where either the microtubular or the nuclear function is impaired to use these single-function variants of this unusual kinesin to link signal-induced changes of cellular architecture to signal-dependent shifts of cellular functions.

## Figures and Tables

**Figure 1 ijms-23-06291-f001:**
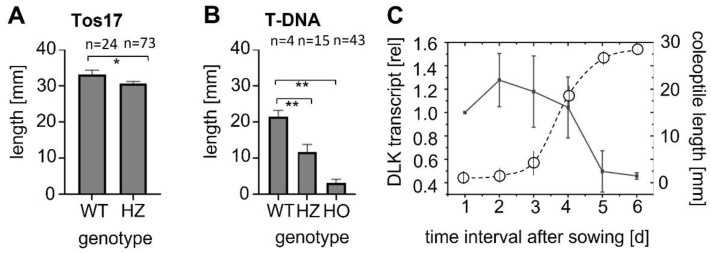
Relationship between DLK expression and coleoptile elongation. Mean length of fully expanded etiolated coleoptiles in the Tos−17 insertion line ND4501_0_508_1A (**A**) and the T-DNA insertion line PFG_3A-07110.R (**B**). WT gives the values for the respective background (‘Nipponbare’ in (**A**), ‘Dongjin’ in (**B**)), HZ for the genotyped heterozygotes, HO for the genotyped homozygotes, and n for the measured seedlings number for each genotype. Error bars represent SE, * significant at *p* < 5%, ** significant at *p* < 1% based on a homoskedastic *t*-test. (**C**) Time course for the steady-state levels of DLK transcripts in etiolated coleoptiles of ‘Nipponbare’ along with a growth curve of ‘Nipponbare’ coleoptiles under these conditions (circles, dotted curve). Each transcript measurement represents the mean from three individuals per experiment, repeated in three technical replicates, and each experiment was repeated in three independent biological series.

**Figure 2 ijms-23-06291-f002:**
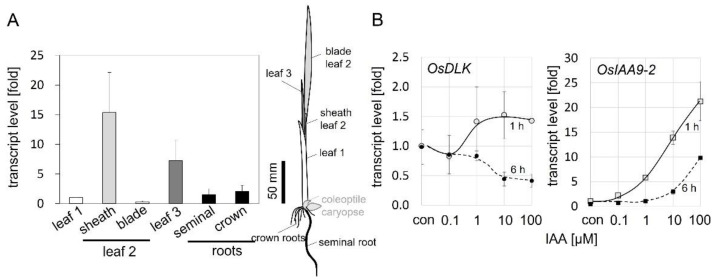
Regulation pattern of OsDLK in seedlings of *O. sativa ssp. japonica* ‘Nipponbare’. (**A**) Steady-state transcript levels of OsDLK in different organs of seedlings raised for 10 days under white light. Values are expressed relative to the level found in the first leaf. The position of these organs is indicated in the schematic drawing. Data represent mean values and SE from at least 12 individuals collected in three different experimental series. (**B**) Dose–response curve for the auxin response of OsDLK in coleoptile segments. Segments of etiolated rice coleoptiles were depleted from endogenous auxin for 1 h and then incubated in different concentrations of indole acetic acid (IAA). As a control for auxin responsivity, transcripts for the auxin-responsive gene *OsIAA9-2* were measured in parallel. Data represent means and SE from three independent experimental series with ten individual segments per measurement.

**Figure 3 ijms-23-06291-f003:**
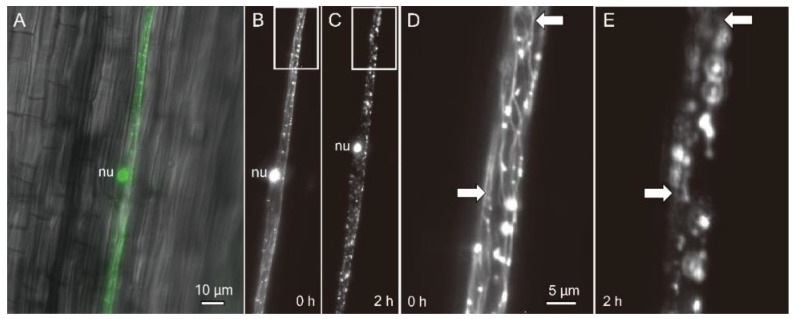
Subcellular localisation of OsDLK in response to cold stress in etiolated coleoptiles of *O. sativa* ssp. *Japonica* ‘Nipponbare’ visualised by a GFP fusion. (**A**) Representative epidermal cell after biolistic transformation (overlay of GFP signal on the differential interference contrast image). (**B**–**E**) OsDLK-GFP signal collected by spinning disc microscopy prior to (**B**,**D**) and 2 h (**C**,**E**) after administering cold stress (0 °C). Survey images (**B**,**C**) and zoom-ins of the region highlighted by the white boxes (**D**,**E**) are shown. White arrows in (**D**,**E**) indicate cortical microtubules (**D**) that are eliminated under cold stress. Note the shift of the nucleus (nu) towards the cell apex concomitant to the elimination of microtubules (**B**,**C**).

**Figure 4 ijms-23-06291-f004:**
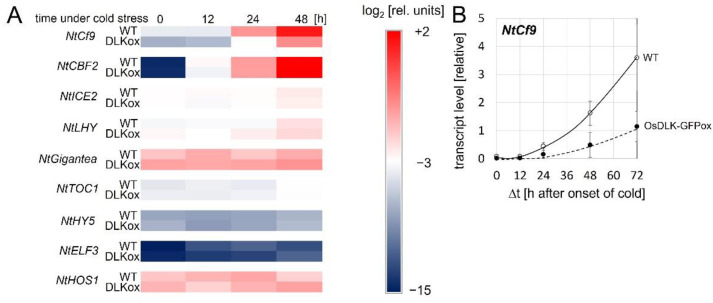
Time course of steady-state transcript levels under continuous cold stress (0 °C) in non-transformed tobacco BY−2 cells (WT), and in cells expressing OsDLK-GFP under control of the CaMV 35S promoter (OsDLK-GFPox). (**A**) Heat map showing transcript levels of NtCf9, the tobacco homologue of CBF4, in comparison to other genes involved in cold signalling. These are Cold Box Factor 2 (CBF2), a transcriptional activator of cold-responsive genes; Inducer of CBF expression (ICE2), the master switch for CBFs; Late Elongated Hypocotyl, a regulator of CBFs acting in parallel with ICE; Gigantea, a positive regulator of freezing tolerance acting independently of CBFs; Timing of Cab Expression 1 (TOC1), a phytochrome-dependent repressor of CBF expression; Hypocotyl 5 (HY5), a light-dependent regulator of cold acclimation acting independently of CBFs; Early Flowering 3 (ELF3), a phytochrome-dependent regulator of CBFs; and High Expression of Osmotically Responsive Genes (HOS1), a negative regulator of CBFs. (**B**) Transcript levels of NtCf9. Data represent means and standard error from five independent experimental series with three technical replications per set. All data are normalised to the same scale, based on the DC_t_ values.

**Figure 5 ijms-23-06291-f005:**
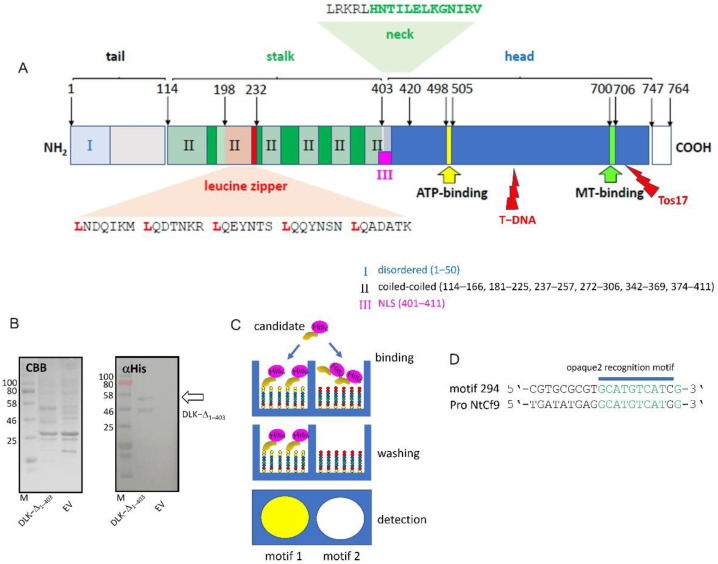
OsDLK qualifies as specific DNA binding protein. (**A**) Domain structure of OsDLK showing the position of the leucine zipper, the nuclear localisation signal (NLS), along with the neck region (the characteristic signature for minus-end-directed motors in bold), the C-terminal motor-head domain, and the ATP-binding and the MT-binding sites. The insertion sites for the mutant lines are given by red lightning bolts. T-DNA indicates the insertion of T-DNA in line 3A-07110 and Tos−17 is the insertion of the Tos−17 retrotransposon in line ND4501. (**B**) Recombinant expression of the N-terminal half (amino acids 1–403, DLK-Δ_1–403_) of DLK versus control cells transformed with the empty vector (EV). The eluents of the Ni-agarose are shown after SDS-PAGE and either staining with Coomassie Brilliant Blue (CBB) or after Western Blotting and probing with αHis antibodies. The respective bands of the expected size are indicated by arrows. (**C**) Principle of DPI-ELISA screening for DNA motives recognised by a DNA-binding candidate protein. Arrays of oligonucleotide motifs coated in microtiter wells are incubated with the His-tagged recombinant candidate. Unbound protein is washed off and bound protein is detected by ELISA. (**D**) High-affinity candidate motif 294 containing an opaque2 recognition motif is found in the promoter of NtAvr9/Cf9.

**Table 1 ijms-23-06291-t001:** Hexanucleotid sequences recovered during the DPI-ELISA assay using recombinantly expressed OsDLKT. NF: No *cis* element was found. The active sites were predicted via PlantCare and highlighted in different colours.

Nr.	Binding Motif	Name	Organism	*cis* Element	Predicted Function
50	TGGTCGATCCGCATGCAGTT	NF	NF	NF	NF
182	GTCTGCGTCCTACCCCATTC	TCA-element	*Brassica oleracea*	GAGAAGAATA	salicylic acid responsiveness
272	GTTCGGGGCTTGGTTTGGAA	ARE	*Zea mays*	TGGTTT	anaerobic induction
294	CGTGCGCGTGCATGTCATCG	O2-site	*Zea mays*	GATGACATGG	regulation of zein metabolism
	CGTGCGCGTGCATGTCATCG	Skn-1_motif	*Oryza sativa*	GTCAT	endosperm expression
299	CTAGGTATCGGTAGGCGCCG	I-box	*Flaveria trinervia*	CCATATCCAAT	anaerobic induction
302	CGCTCCGTTTTTGCAATGCG	CAAT-box	*Hordeum vulgare*	CAAT	regulation of zein metabolism

## Data Availability

All data supporting the findings of this study are available within the paper and within its Appendix A published online.

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
