# Peer review of "The Minus-End-Directed Kinesin OsDLK Shuttles to the Nucleus and Modulates the Expression of Cold-Box Factor 4"

_ijms, 2022, doi:10.3390/ijms23116291_

Round 1

Reviewer 1 Report

The manuscript describes the biological function of the kinesin motor DLK of rice, by analysis of the phenotype of DLK mutants, DLK expression in several plant tissues, a DPI-ELISA and its subcellular localization in coleoptiles under cold stress. The study is interesting and is well explained. However, several suggestions are made to improve the manuscript.

Line 115 – The authors wrote the sentence: “Relationship between DLK and coleoptile elongation”. Do you meant to write "Relationship between DLK expression and..."?

Line 119 – Do authors intend to write the term homoscedastic?

Line 117 - Please revise the position of the single quotation marks for the Nipponbare and Dongjin cultivar names in Fig. 1, Fig. 2, Fig.3 and along the text.

Line 138 – Do you mean that limited expression of OsDLK or its absence impairs coleoptile elongation?

Line 155 – The “Dose-response curve” is not for assessing the auxin response, but for assessing the synthesis of OsDLK transcripts. Please revise.

Check out the following sentence: ”Segments of… were depleted…”.

Fig. 2B. What is written in the Y axis legend?

Line 171 – Authors refer that the slope of the curve when OsIAA9-2 expression is under analysis (Fig. 2B) was comparable to that observed 1h after addition of IAA. The sentence is confuse, so please explain what you mean.

Line 404 – Do you mean that “cold hardening is accompanied by the high expression of cold box factor 4”?

Material and Methods section

Line 494 – I understood that pK7FWG2-OsDLK has a GFP sequence fused with OsDLK. In this case, the construction should be clarified in the text. It is not the plasmid that is “transiently transformed”, but the rice coleoptiles that are transformed.

Line 542 – The description “quantitative reverse transcription-polymerase chain reaction” should be altered to “quantitative Real-time polymerase chain reaction”. When reverse transcription is performed, the enzyme reverse transcriptase is used in the PCR mix, which is not the case.

Line 554 – Please explain how exogeneous OsDLK transcripts were added to tobacco cell line BY-2. What was the methodology used? or had the cell lines already been transformed?

Line 560 – the authors describe distinct temperatures for the hybridization and polimerization step of a semi-quantitative real-time PCR. It seems like a PCR, not a real-time PCR. Please explain the procedure

Line 563 – authors wrote that quantitative RT-PCR performed on the tobacco BY-2 cells was conducted as described for rice coleoptiles. Which were the reference genes used to quantify gene expression in tobacco cell lines?

Line 576 – Please confirm the final molarity of IPTG in bacterial culture, was 200 nM or 200 mM?

Please revise the "references" section, the journal's abbreviated name is missing from all references.  Please pay attention too to the scientific names of species that should be written in italics.

Reviewer 2 Report

The authors characterized dynamics of transcription and subcellular localization of OsDLK in rice and performed phenotypic analysis of mutants of OsDLK. The authors also found potential downstream genes regulated by OsDLK by DPI-ELISA screening. There are several points should be addressed as following,

The T-DNA allele showed more dramatic phenotype than the Tos-17 allele even though the homozygous T-DNA allele can survive unlike Tos-17 allele. Please describe Tos-17 and T-DNA insertion sites and discuss this. Please also provide information of seed size of the alleles to rule out a possibility that seed development, especially endosperm development, had been already altered by the mutations.

According to mRNA levels of OsDLK at a seedling stage, OsDLK is expressed in sheath at the highest level. Authors should report phenotypes of the heterozygous and homozygous plants at seedling or later stages. This will be important information for readers.

I understand that cold stress likely induces detachment of OsDLK-GFP from the microtubule, but there is still a possibility that microtubule itself is disassembled in rice in this condition. It is needed to distinguish them by additional experiments such as monitoring tubulin fused with a FP or co-treatment with taxol which prevents microtubule disassembly. Method of microscopy with chilling is not well described but if the etiolated coleoptiles were mounted under coverslips during the cold treatment, mechanical pressure might also affect the microtubule assembly. If so, control experiments without cold stress should be needed.

Minor points,

  • Please describe information about black line piercing the boxes (it appears in before ATG), blue boxes, region of 747–764, and where the stop codon is, in Figure 5A. This domain structure illustration is quite complicated. Please stretch this image sideways.
  • What are N=97 and N=62 in Figure 1A and B? Did the authors use same number of seedlings for each genotype? Please describe this in the legend.

  • Do the plots of coleoptile length have error bars? The circle is too large to show error bars. Please modify this.

  • Value of Y axis is missing in Figure 4B.

  • L290: Sentence should be revised, no cis element was found?

Round 2

Reviewer 2 Report

I appreciate the authors' effort to improve the manuscript. It is now satisfying. One thing, in Figure 4B, description of Y axis is odd; [ ] ().

Author Response

Comments and Suggestions for Authors

I appreciate the authors' effort to improve the manuscript. It is now satisfying. One thing, in Figure 4B, description of Y axis is odd; [ ] ().

Response: We thank the reviewer for the comment and we are sorry for the mistake. The bracket of Figure 4B has been modified as "transcript level [fold]".